# NNC 26-9100 increases Aβ1-42 phagocytosis, inhibits nitric oxide production and decreases calcium in BV2 microglia cells

**Joseph Schober** *, **Jahnavi Polina, Field Walters, Nathan Scott, Eric Lodholz, Albert Crider, Karin Sandoval, Ken Witt**

Department of Pharmaceutical Sciences, School of Pharmacy, Southern Illinois University Edwardsville, Edwardsville, Illinois, United States of America

* joschob@siue.edu

**Data Availability Statement:** All relevant data are within the paper main figure sequence. There is no Supporting or Supplemental data in this submission.

## Abstract

Microglia are the resident immune cell of the brain involved in the development and progression of Alzheimer's disease (AD). Modulation of microglia activity represents a potential mechanism for treating AD. Herein, the compound NNC 26–9100 (NNC) was evaluated in toxicity, nitric oxide release, $A\beta_{1-42}$ uptake and cytosolic calcium assays during lipopolysaccharide (LPS)-activated conditions using mouse BV2 microglia cells. After 24 hours, LPS increased cell toxicity in the alamar blue and lactate dehydrogenase assays, increased nitrite release, and increase cytoplasmic calcium. Addition of NNC decreased the LPS-induce lactate dehydrogenase release, had no effect in the alamar blue assay, decreased nitrite release and decreased cytosolic calcium. In the absence of LPS, NNC increased uptake of FITC-tagged $A\beta_{1-42}$. These data demonstrate that NNC treatment decreases nitrosative stress and microglia cell damage during LPS-induced activation and enhances phagocytosis of $A\beta_{1-42}$ during non-inflammatory conditions. Thus, NNC 26–9100 may have beneficial effects in AD and in inflammatory diseases of the brain through enhancement of microglial Aβ clearance, and cell protective effects through prevention of elevated cytosolic calcium and inhibition of nitric oxide release.

## Introduction

As the resident immune cells of the central nervous system (CNS), microglia play an essential role in the maintenance and function of the brain. The purpose of microglia is to defend the CNS against injury, contribute to repair, and support neuronal networks. Microglia involvement in neurodegenerative disease has long been the subject of research, with both positive and negative implications [1]. Alzheimer's disease (AD) exemplifies the dualistic nature of microglia [2]. In AD, amyloid-beta peptide (Aβ) accumulates within the brain, ultimately resulting in Aβ plaques [3]. Microglia upregulate and accumulate around these plaques [1, 4], with the capacity to recognize and phagocytize Aβ via a number of cell surface receptors [5]. Several studies have shown microglia to reduce Aβ levels and halt pathogenesis [6]. However,

**Funding:** This research was supported by the National Institute on Aging of the National Institutes of Health, grant R01AG047858. Somataolynk Inc. did not provide financial support for the study, and did not play a role in the study design, data collection, data analysis, decision to publish, or preparation of the manuscript.

**Competing interests:** Authors K. Witt, A. Crider, and K. Sandoval are co-inventors on patent applications for the development of SSTR4 agonists, through Southern Illinois University Edwardsville (USA patent application No. PCT/US2018/032368; European National Application No. 18798793.8, Japanese Application No. 2019-562574). Ken Witt is CEO of Somatolynk Inc., a company focused on advancement of SST4 agonists. These patent applications and the commercial affiliation with Somatolynk Inc. does not alter our adherence to PLOS ONE policies on sharing data and materials.

prolonged microglial activation can also impaired Aβ degradation, contributing to neuronal synaptic loss and downstream enhancement of tau pathology [5]. The point and circumstances in which microglia transition from being beneficial to harmful remains unclear. This has much to do with the highly dynamic nature of microglia, individual genetic variables, and remaining uncertainties of AD pathology. Nevertheless, microglia represent a viable target for AD and other inflammatory processes in the brain.

Our previous studies have shown the selective $SSTR_4$ (somatostatin receptor type 4) agonist, NNC 26–9100 (NNC), enhanced learning and memory in mouse models of AD, increased activity of the Aβ degrading enzyme neprilysin, and decreased protein expression of Aβ-oligomers [7–9]. We recently evaluated the effects of NNC administration in the 3xTg mouse model of AD, identifying microglia-associated changes [10]. NNC increased cortical mRNA expression of the Aβ-degrading enzymes *neprilysin* (9.3-fold) and *insulin degrading enzyme* (14.8-fold). NNC also decreased cortical expression of *cluster of differentiation-33 (Cd33)* by 25%, while increasing cortical and subcortical *macrophage scavenger receptor-1 (Msr1)* by 1.8 and 2.0-fold, respectively. Microglia secretion of insulin degrading enzyme has been previously shown to be regulated by SRIF (somatotropin release-inhibiting factor) [11]. Additionally, *Cd33* downregulation and *Msr1* upregulation correspond with microglia-associated Aβ phagocytosis [12, 13]. While data suggest that NNC is regulating microglia activity in whole brain tissue, isolated cellular responses have not been characterized. Here, we evaluated NNC treatment of BV2 microglia in response to LPS-induced inflammation in cellular viability, nitrite output, $Aβ_{1-42}$ uptake, and intracellular calcium assays. The timeframe of BV2 experiments were based on previous *in vivo/ex vivo* evaluations in AD mouse models that identified Aβ degradation after NNC treatment over a similar period [7, 8, 10].

## Materials and methods

### Chemicals and reagents

All chemicals obtained from Sigma-Aldrich (St. Louis, MO), unless otherwise stated. Dulbecco's Modified Eagle's Medium (DMEM), phosphate-buffered saline (PBS, without calcium and magnesium) 0.05% Trypsin/0.53 mM ethylenediaminetetraacetic acid (EDTA), and heat-inactivated fetal bovine serum (FBS) obtained from Corning Life Sciences (Manassas, VA). The fluorescein isothiocyanate (FITC) tagged $Aβ_{1-42}$ monomer peptide (FITC-β-Ala-Amyloid β-Protein (1–42) ammonium salt) obtained from Bachem (Torrance, CA). The calcium probe, Fluo-8 AM, was purchased from AAT Bioquest (Sunnyvale, CA). NNC was synthesized, purified, and confirmed via MMR by Dr. A. Crider per previously established protocols [14, 15].

### Cell culture and treatment

BV2 cells, generated from retrovirus immortalization of mouse microglia cells [16], were used as they preserve primary morphological, phenotypical, and functional properties with maintenance of inflammatory activation [16, 17], and express $SSTR_4$ [18]. The BV2 cells were maintained in growth media (DMEM, 1% penicillin-streptomycin-amphotericin B and 10% FBS) at 37°C, 5% $CO_2$ incubator in 25 $cm^2$ flasks until 90% confluent. Trypsinized cells were suspended in growth media containing 0.5% FBS and count was adjusted to 200 cell/μL. In all assays and experiments, 312.5 μL of cell suspension was added per $cm^2$ to maintain a constant density of 62,500 cells/$cm^2$. At 30 min post-plating, NNC and LPS were added in a volume equal to 50% of the cell suspension volume to maintain consistent media depth.

## Alamar blue assay

Alamar Blue is a cell viability assay that provides a rapid and sensitive measure based on reduction capacity of living cells. The assay was performed according to manufacturer procedures (Thermo Fisher Scientific, Waltham, MA). Following 24 hr at 37˚C, 5% CO2, media was removed for use in other assays, and 96-well plates were washed twice with 100 μL PBS and 100 μL fresh cell growth media washed added to each well. 10 μL of alamar blue reagent was added to the media of each well and the plate was placed in the 37˚C, 5% $CO_2$ incubator for 2 hr before reading fluorescence at excitation 544 nm and emission 490 nm via PolarStar Omega (BMG Labtech, Cary, NC., USA).

## Lactate dehydrogenase assay

LDH is a cytosolic enzyme released into the culture media when the plasma membrane is permeable and serves as an indicator of cellular membrane integrity. The assay was performed according to manufacturer procedures (Thermo Fisher Scientific). Briefly, 50 μL of cell culture media per well was taken after 24 hr treatment as described above, and transferred to a new 96-well microplate. 50 μL of reaction mixture was added to each well, incubated for 30 min, followed by stop solution. The reaction product was quantified by absorbance at 492 nm and 690 μm via Multiskan (Thermofisher) plate reader.

## Nitrite assay

Nitrite serves as a stable surrogate of nitric oxide (NO), a critical effector molecule associated with microglia activation and nitrosative stress. The Nitrite Measure-iT assay performed according to manufacturer procedures (Thermo Fisher Scientific). Briefly, 10 μL of cell culture media per well taken after 24 hr treatment as described above, and transferred to a new 96-well microplate. 100 μL of the working solution was added to each well, incubated at room temperature for 10 min, and followed by 5 μL of quantitation developer. Supplied nitrite standards followed identical procedure. Measurement of fluorescence at excitation 365 nm and emission 410 nm via a PolarStar Omega.

## Sytox green, Aβ1–42 uptake, SSC, FSC, and cytosolic calcium

Flow-cytometry evaluations followed NNC and LPS treatment of 24 hours after plating. For cell viability measurements, NNC and LPS treatment was followed by sytox green staining for an additional 30 min. Sytox green is a high-affinity nucleic acid stain that easily penetrates cells with compromised plasma membranes, but will not penetrate healthy cell membranes. Wells were washed once with PBS and the cells were removed by 2 min incubation with a 0.05% Trypsin/0.53 mM EDTA solution at 37˚C and then mixed with 0.5 mL media containing 10% FBS. Additionally, the cell population was gated for side-scattering (SSC) versus forward-scattering (FSC) height plots, and the mean fluorescence intensities within the gate was measured. For $Aβ_{1-42}$ uptake determinations, 100 nM FITC-$Aβ_{1-42}$ was added to the media, incubated with NNC and LPS for 24 hr, and then assessed by flow-cytometry. The 2 hr treatment was added at 22 hr of NNC/LPS treatment, with maintenance of NNC/LPS up to the 24 hr time point. Samples were analyzed using an Accuri C6 flow cytometer (BD Biosciences, San Jose, CA, USA). The BV2 cell population was gated and the mean fluorescence intensities within the gate were measured for FITC-$Aβ_{1-42}$ peptide. For cytosolic calcium measurements, 2 μM Fluo-8 AM was added for the last 30 min of the 24 hr treatment period. Cells were removed by 2 min incubation with 0.1% Trypsin (without EDTA) and analyzed by flow cytometry. The concentration of Fluo-8 AM in the cell suspension was maintained at 2 μM.

Raw fluorescence values were converted to nM calcium using the following formula and Kd value recommended by the manufacturer. [Ca] is the cytosolic calcium concentration in nM, F is the treatment fluorescence, Fmin is the background fluorescence, Fmax is the maximal fluorescence at probe saturation determined by brief treatment with 0.01% Triton-X 100, and Kd is the dissociation constant of the probe = 389 nM.

$$[Ca] \; = \; Kd(F - Fmin)/(Fmax - F)$$

## Confocal microscopy

Microscopy was performed to confirm intracellular localization of FITC-A$\beta_{1-42}$. Glass coverslips were coated with 50 μg/mL fibronectin for 20h at 4˚C. 3 mL of cell suspension was added onto the fibronectin-coated coverslips in 35 mm dishes and incubated for 30 min in a 37˚C, 5% $CO_2$ incubator followed by addition of 100 nM A$\beta_{1-42}$ peptide. After incubation for 24 hours, the coverslips were fixed in buffer containing 4% para-formaldehyde and 0.1% Triton X-100 in PBS. Coverslips were washed in deionized water and then blocked with 2% BSA solution in PBS for 20 min at room temperature. Samples were incubated with Hoechst 33258 (DNA/nuclear stain) and Alexa Flour-546-phalloidin (Actin stain) for 20 min at 37˚C, washed in deionized water and then mounted on glass slides using Aqua-Poly/Mount. Z-series were acquired using an Olympus FV1000 laser scanning confocal microscope equipped with a 1.42 NA 60X objective lens (Center Valley, PA, USA). Using Metamorph software (San Jose, CA), the raw image stacks were rescaled, converted to 24-bit and de-noised with a low pass filter before 3D reconstruction. Images were rescaled and color combined in Adobe Photoshop (San Jose, CA).

## Statistical analysis

All observations are shown in plots as dots, source data summated across all experiments and provided in supplemental materials. The significance level was set at α = 0.05 for all analyses. A two-way design was employed for all analyses utilizing robust statistics through the R package, WRS2 [19]. Two-way robust ANOVAs were performed based on the estimator of location (20% trimmed means, modified one-step M-estimator, or ranks [20]). For 20%-trimmed means, a bootstrap t method was used. For the modified one-step estimator, a percentile bootstrap was used. Multiple comparison tests were conducted in a similar manner for bootstrapping with p-values adjusted for the family wise error rate using the Bonferroni correction. For ranks, the robust analog of Cohen's d was utilized using the median and the percentage bend midvariance.

## Results

### Alamar blue

Alamar blue was used to measure changes in cell viability of BV2 cells in response to LPS (0, 0.83 and 8.3 ng/ml) and NNC (0, 0.104, 0.208, 0.417 and 0.833 μM). Alamar blue reduction was only impacted by LPS (Fig 1). As the concentration of LPS increased, reduction of alamar blue decreased, with significance found between all concentrations of LPS (p<0.0001 between each concentration, Bonferroni).

### Lactate dehydrogenase

Membrane integrity of BV2 cells was next assessed in response to LPS and NNC using lactate dehydrogenase (LDH). Both LPS and NNC significantly affected LDH (Fig 2). As LPS

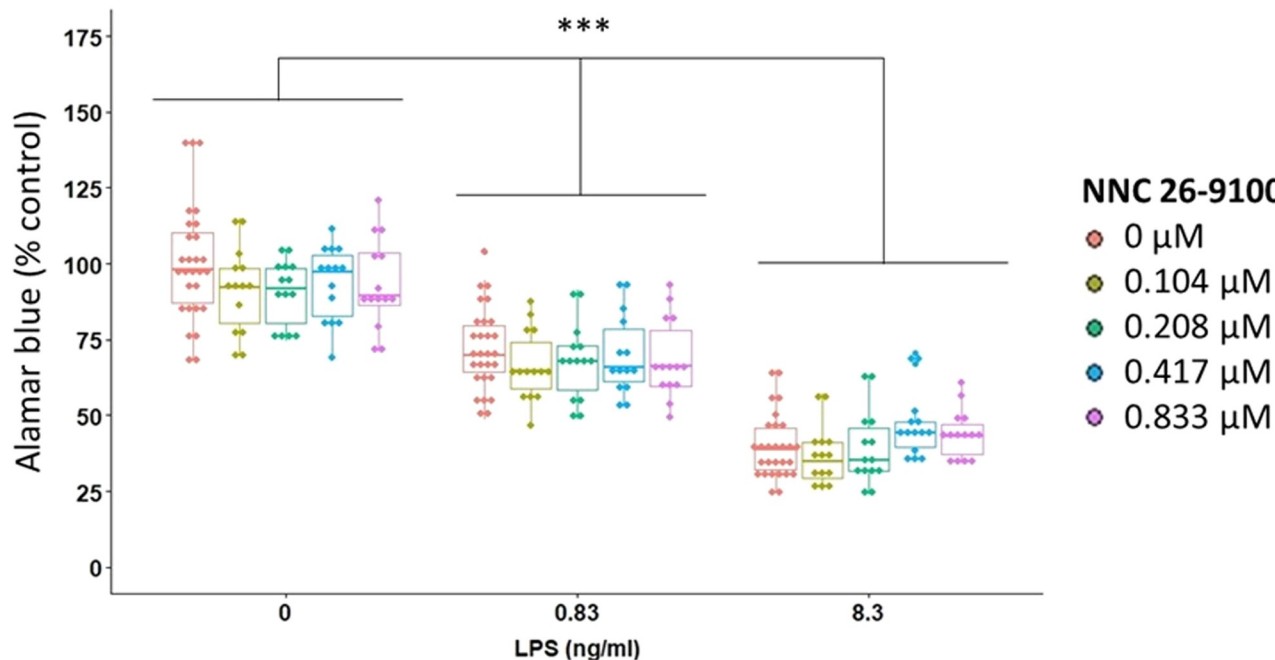

**Fig 1. NNC Effect on alamar blue assay.** NNC treatment against LPS at 24 hr (n = 14/group; LPS: p<0.0001; NNC: p = 0.101; NNC*LPS: p = 0.0885, Two-way robust ANOVA using 20% trimmed means). ***p<0.0001, Bonferroni post-hoc test.

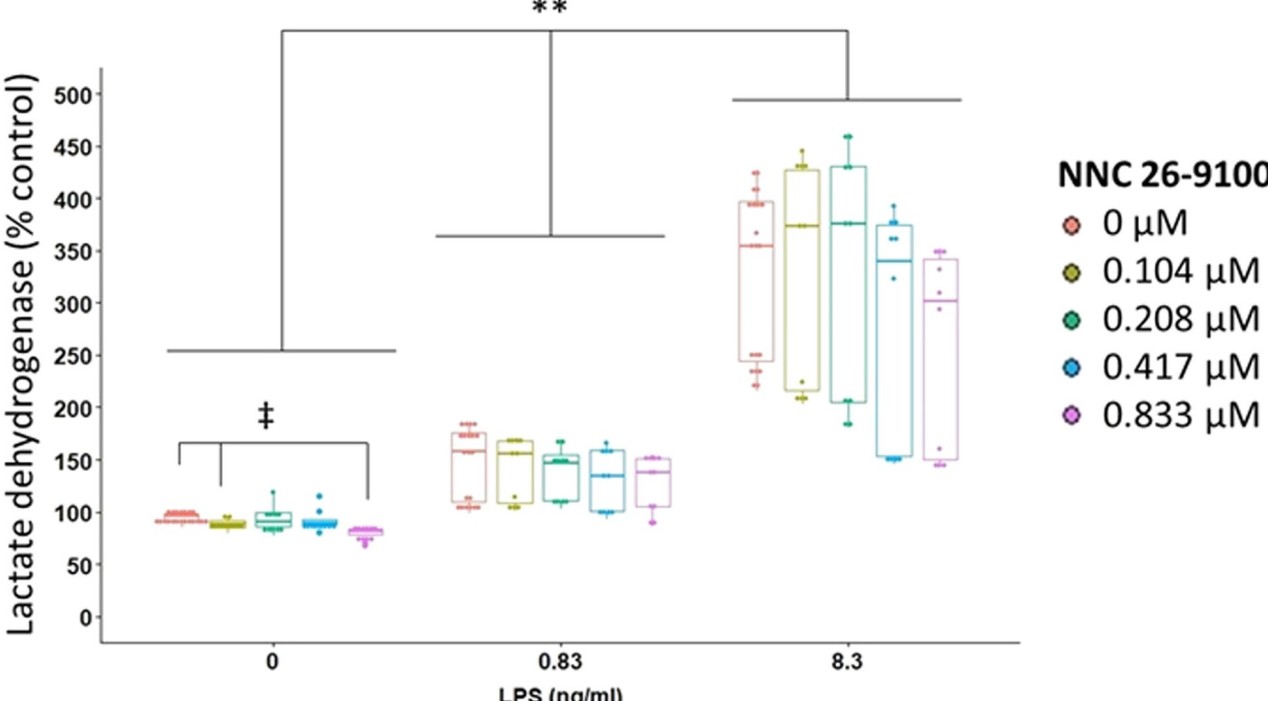

**Fig 2. NNC effect on LDH assay.** LDH determination of NNC treatment against LPS stimulation at 24 hr (n = 10/group; LPS: p<0.0001; NNC: p<0.0001; NNC*LPS: p = 0.4520, Two-way rank based ANOVA). Subgroup analyses of LDH with NNC within each concentration of LPS (0 ng/ml LPS: p = 0.001, 0.83 ng/ml LPS: p = 0.23, 8.3 ng/ml LPS: p = 0.12, One-way rank based ANOVA). **p<0.001, ‡ p<0.05, Cliff's Wilcoxon Mann-Whitney with Bonferroni adjustment.

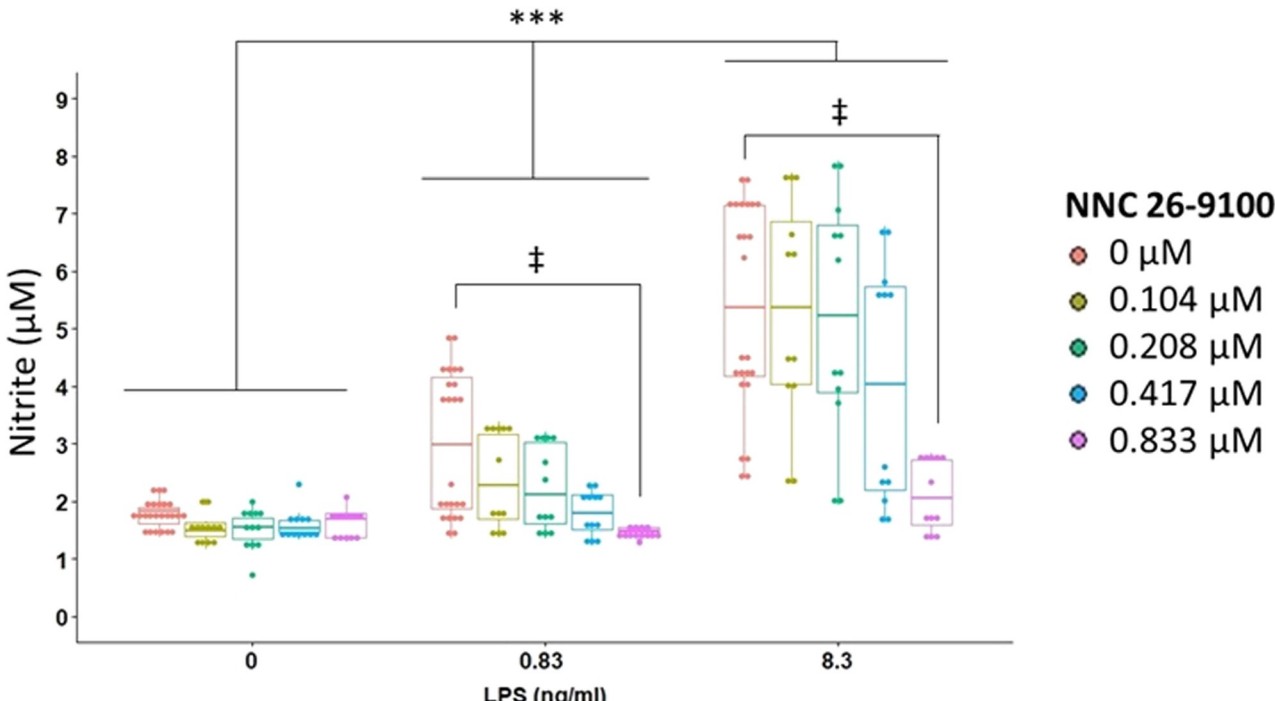

**Fig 3. NNC effect on nitrite output.** NNC treatment against LPS stimulation at 24 hr (n = 12/data point; LPS: p<0.0001, NNC: p<0.0001, NNC*LPS: p = 0.0024, Two-way rank based ANOVA) ***p<0.0001, ‡ p<0.05, Cliff's Wilcoxon Mann-Whitney with Bonferroni adjustment.

increased, the percent LDH relative to control also increased, with significant differences found between all LPS concentrations (p = 0.0003 between each LPS concentration, Cliff's WMW, Bonferroni).

Subgroup analyses were performed within each concentration of LPS to determine whether NNC significantly influenced LDH. Within the 0 ng/ml LPS concentration, LDH was significantly impacted by NNC. Within the 0 ng/ml LPS concentration, as the concentration of NNC increased, LDH decreased becoming significant when comparing 0.833 μM to 0.104 μM NNC, and when comparing 0.833 μM to 0 μM NNC (p = 0.034 and p = 0.0017, respectively, Cliff's WMW, Bonferroni).

### Nitrite

Microglial nitrite output by BV2 cells was next assessed in response to LPS and NNC (Fig 3). A significant interaction was found between NNC and LPS on nitrite. Within 0 μM NNC, as the concentration of LPS increased, nitrite increased, with significant differences found between all LPS concentrations (p = 0.018 between each LPS concentration, CWMW, Bonferroni). Within 0.833 μM NNC, LPS nitrite levels were found to be similar between all concentrations of LPS.

Within 0 ng/ml LPS, nitrite levels were similar between all concentrations of NNC. Within both 0.83 ng/ml LPS and 8.3 ng/ml LPS, nitrite levels significantly decreased when comparing 0.833 μM to 0 μM NNC (p = 0.0015 for 0.83 and 8.3ng/ml LPS, CMWM, Bonferroni). Data support the ability of NNC to reduce the LPS-induced NO output.

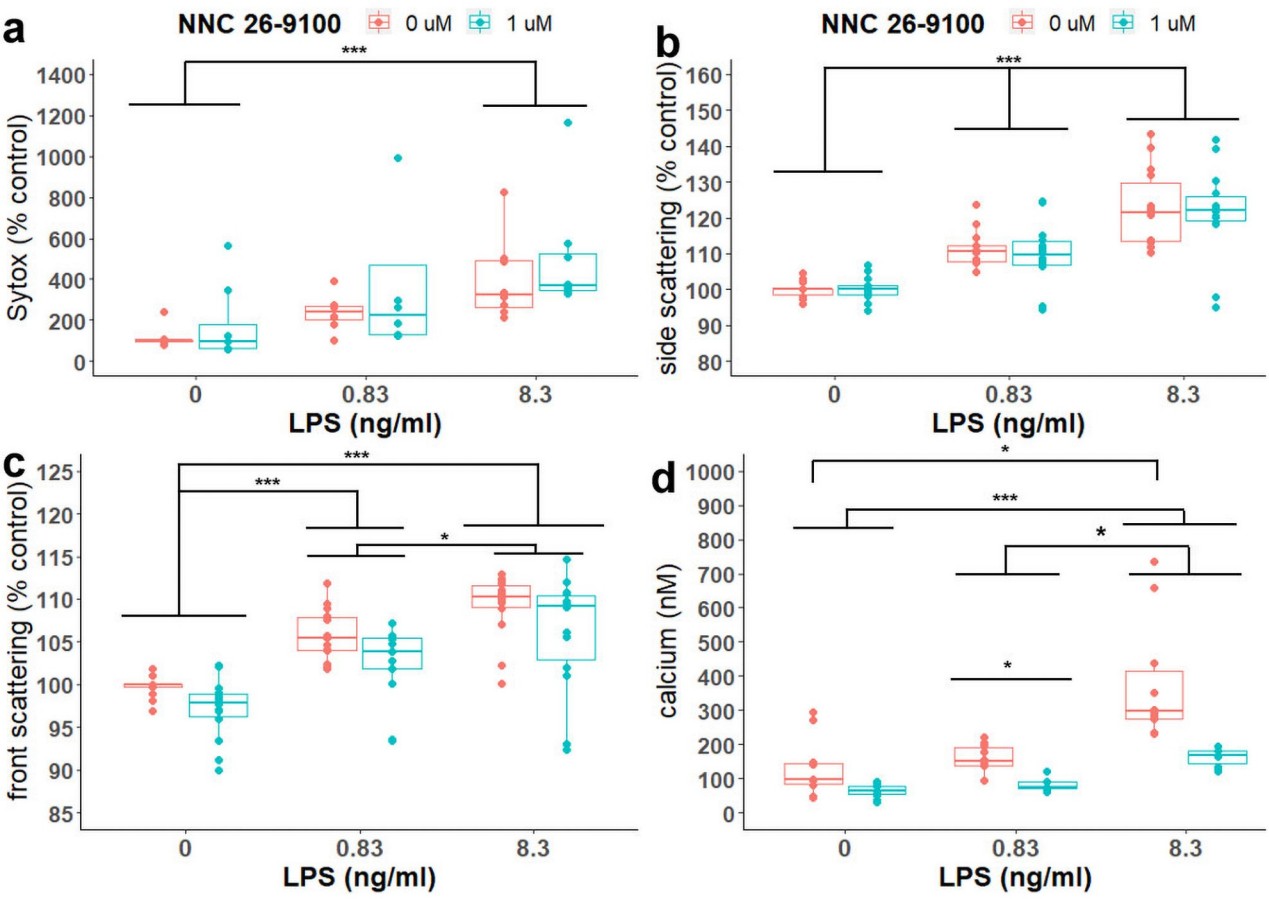

**Fig 4. Effects of NNC on sytox, cell scattering and intracellular calcium.** Flow-cytometry evaluation of NNC evaluated against LPS stimulation on **(A)** sytox (n = 8 /group; LPS: p<0.0001; NNC: p = 0.186; LPS*NNC: p = 0.87, Two-way rank based ANOVA), Bonferroni post-hoc test or Cliff's Wilcoxon Mann-Whitney with Bonferroni adjustment, respectively. **(B)** side-scattering (SSC, n = 28/data point; LPS: p<0.0001, NNC: p = 0.8875; NNC*LPS: p = 0.995) and **(C)** forward scatter (FSC; LPS: p<0.0001; NNC: p = 0.0715, NNC*LPS: p = 0.9825), Two-way robust ANOVA. Bonferroni post hoc and **(D)** intracellular calcium (NNC: p<0.0001, LPS: P<0.001, NNC x LPS: p >0.5. Main effect post-hoc analyses are not shown. Two-way robust ANOVA. Bonferroni post-hoc). ***p<0.0001, **p<0.01, *p<0.05.

## Sytox cell viability

The impact of NNC (0 or 1 μM) against LPS (0, 0.83, and 8.3 ng/ml) treatment was assessed at 24 hours on sytox green (Fig 4a). LPS significantly influenced sytox green, with increased sytox green as the concentration of LPS increased. Sytox was significantly higher when comparing both 0.83 and 8.3 ng/ml LPS to 0 ng/ml LPS (p = 0.006 and 0.003, respectively, CWF) and comparing 8.3 ng/ml LPS to 0.83 ng/ml LPS (p = 0.027, CWF).

## SSC and FSC

NNC (0 or 1 μM) with LPS (0, 0.83, and 8.3 ng/ml) treatment was evaluated at 24 hours on side-scattering (SSC) and front-scattering (FSC) in conjunction with FITC-Aβ1–42. LPS significantly influenced both SSC (Fig 4b) and FSC (Fig 4c).

LPS dose-dependently increased both the percent SSC and FSC. Both SSC and FSC were significantly higher when comparing 0.83 ng/ml LPS to 0 ng/ml LPS (SSC: p<0.0001, FSC: p<0.0001), 8.3 ng/ml LPS to 0.83 ng/ml LPS (SSC: p<0.0001, FSC: p<0.0001, Bonferroni), and 8.3 ng/ml to 0 ng/ml LPS (SSC: p<0.0001, FSC: p = 0.006, Bonferroni).

## Calcium

We next tested the dose-dependent effects of NNC and LPS on calcium in BV2 cells (Fig 4d). Both NNC and LPS treatment independently influenced calcium (NNC: p<0.0001, LPS: P<0.001, NNC x LPS: p >0.5). Across all cells, calcium levels were significantly lower with 1 μM NNC compared to vehicle (P<0.001, main-effect of NNC) and significantly increased as LPS increased (Main effect of LPS, 0 vs 0.83: p = 0.0465, 0 vs 8.3: p<0.0001, 0.83 vs 8.3 p = 0.0045, Bonferroni).

Within vehicle treated cells, calcium levels significantly increased at 8.3 ng/ml LPS compared to 0 ng/ml LPS (p = 0.0015, Bonferroni). Within NNC treated cells, calcium levels were significantly higher when comparing 8.3 ng/ml LPS to 0 ng/ml LPS (p<0.0001, Bonferroni) and when comparing 8.3 ng/ml LPS to 0.83 ng/ml LPS (p = 0.0035, Bonferroni). When examining the effect of NNC on calcium within each LPS concentration, 1 μM NNC significantly lowered calcium levels when compared to vehicle within the 0.83 ng/ml LPS concentration (p = 0.0035, Bonferroni). While calcium levels were lower with 1 μM NNC compared to vehicle within the two other LPS concentrations, they were not significant.

## FITC-A$\beta_{1-42}$ uptake

The impact of NNC and LPS on the uptake of FITC-A$\beta_{1-42}$ by BV2 microglia was assessed over 24 hr of treatment (Fig 5a), as well as during the final 2 hr of the 24 hr treatment (Fig 5b). At 24 hr, both LPS and NNC independently affected FITC-A$\beta_{1-42}$ uptake by BV2 cells. FITC A$\beta_{1-42}$ uptake increased as the concentration of LPS increased, with significant differences found between all concentrations of LPS (p<0.0001 between each concentration of LPS, Bonferroni). NNC at 1 μM significantly increased FITC-A$\beta_{1-42}$ uptake compared to 0 μM NNC (p<0.0001, Bonferroni). Subgroup analyses were next performed comparing FITC-A$\beta_{1-42}$uptake between 1 and 0 μM NNC within each concentration of LPS. Within only 0 ng/ml LPS, NNC significantly increased uptake of FITC-A$\beta_{1-42}$ compared to 0 μM NNC (p<0.0001, Bonferroni). Within 0.83 ng/ml and 8.3 ng/ml LPS, uptake of FITC-A$\beta_{1-42}$ was similar between NNC concentrations.

With the 2 hr FITC A$\beta_{1-42}$ uptake, LPS treatment significantly affected FITC A$\beta_{1-42}$ uptake. As LPS increased, FITC A$\beta_{1-42}$ uptake increased with significant differences found between all concentrations of ng/ml LPS (p<0.0001 between each LPS concentration, Bonferroni). Subgroup analyses were performed looking at FITC-A$\beta_{1-42}$ uptake between 1 and 0 μM NNC within each concentration of LPS. Within 0 ng/ml LPS, 1 μM NNC significantly increased uptake of FITC-A$\beta_{1-42}$ uptake compared to 0 μM NNC (p<0.0001, Bonferroni). Uptake of FITC-A$\beta_{1-42}$ between NNC concentrations was similar within 0.83 ng/ml and 8.3 ng/ml LPS. The 24 hr and 2 hr FITC A$\beta_{1-42}$ had similar trend, supporting SSTR$_4$ agonist mediated uptake under non-LPS conditions. In confocal experiments, we confirmed that the cell-associated FITC-A$\beta_{1-42}$ was indeed located intracellularly in the BV2 cells. As shown in the representative images, FITC-A$\beta_{1-42}$ was observed throughout the cell thickness (Fig 6a) in a perinuclear pattern (Fig 6b) consistent with localization to late endosomes or lysosomes [21].

## Discussion

Correlation of brain SRIF decreases with AD progression and other neurological disease has been observed for a number of years throughout multiple studies [22]. Decline of signaling through one or more SSTR receptor subtypes throughout the brain of AD patients [23] is likely central to disease processes including plaque formation [22] and inflammation [24]. Microglia involvement in inflammatory processes and their capacity to phagocytize A$\beta$ identifies a central role in the progression of AD [25]. In this study, we evaluated the effect of a SSTR$_4$ agonist,

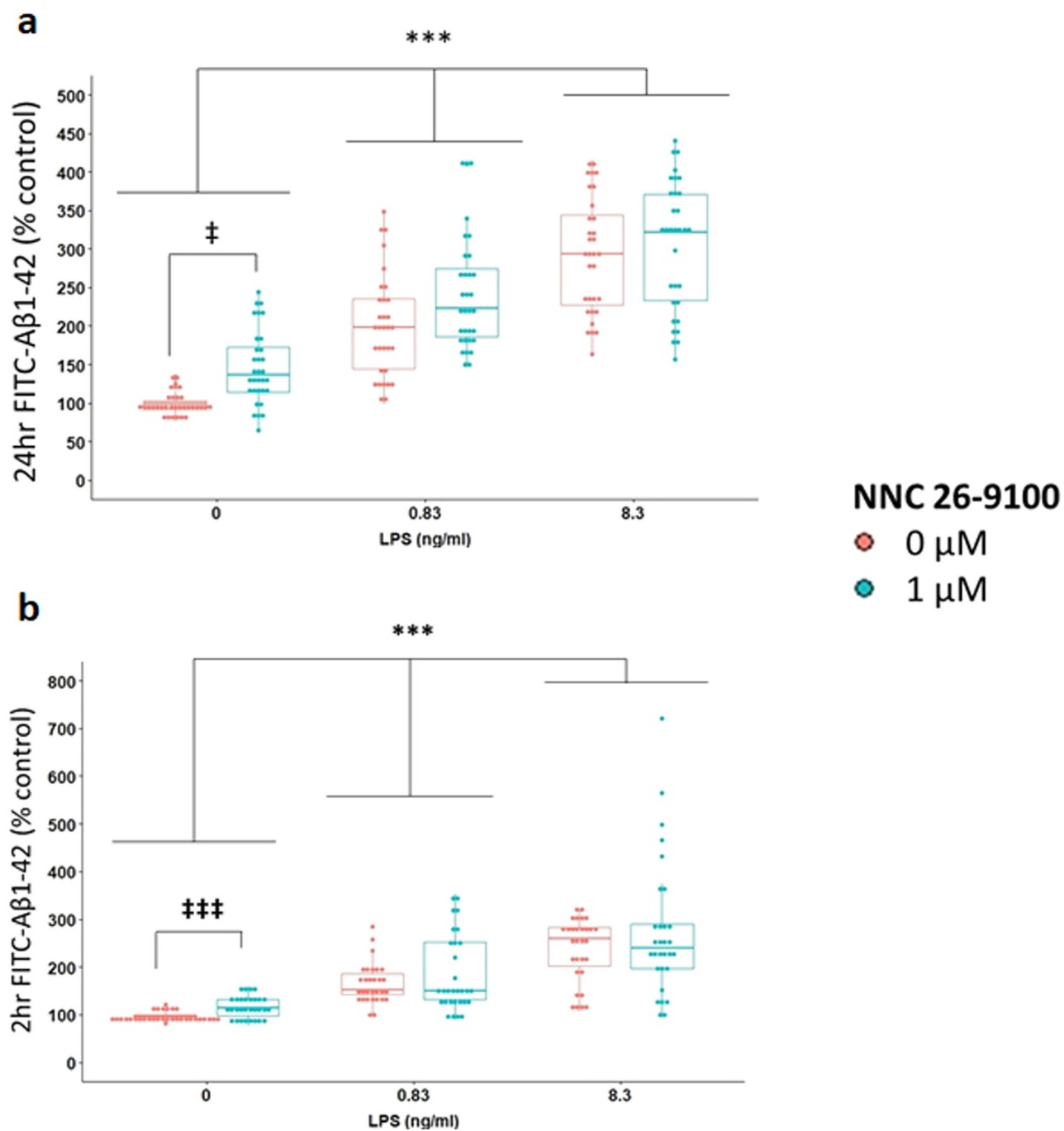

**Fig 5. Effect of NNC on Aβ$_{1-42}$ uptake. (A)** 24 hr FITC-Aβ$_{1-42}$ uptake of NNC treatment against LPS stimulation (n = 32/data point). LPS: p<0.0001, NNC: p = 0.0105, NNC: LPS*NNC: p = 0.414, Two-way robust ANOVA, ***p<0.0001, ‡p<0.05, Bonferroni. **(B)** 2 hr FITC-Aβ$_{1-42}$ uptake of NNC against LPS stimulation (n = 30/data point). LPS, p<0.0001, NNC: p = 0.5755, NNC*LPS: p = 0.228, Two-way robust ANOVA. ‡‡‡p<0.0001, ***p<0.0001, Bonferroni.

NNC 26–9100, on BV2 microglia response to LPS-induced inflammation. LPS has shown to rapidly activate microglia with corresponding changes in morphology, with increased inflammatory mediator release and phagocytosis affiliated with the M1 phenotype [26]. While LPS decreased microglia viability as measured by alamar blue in our examination, consistent with

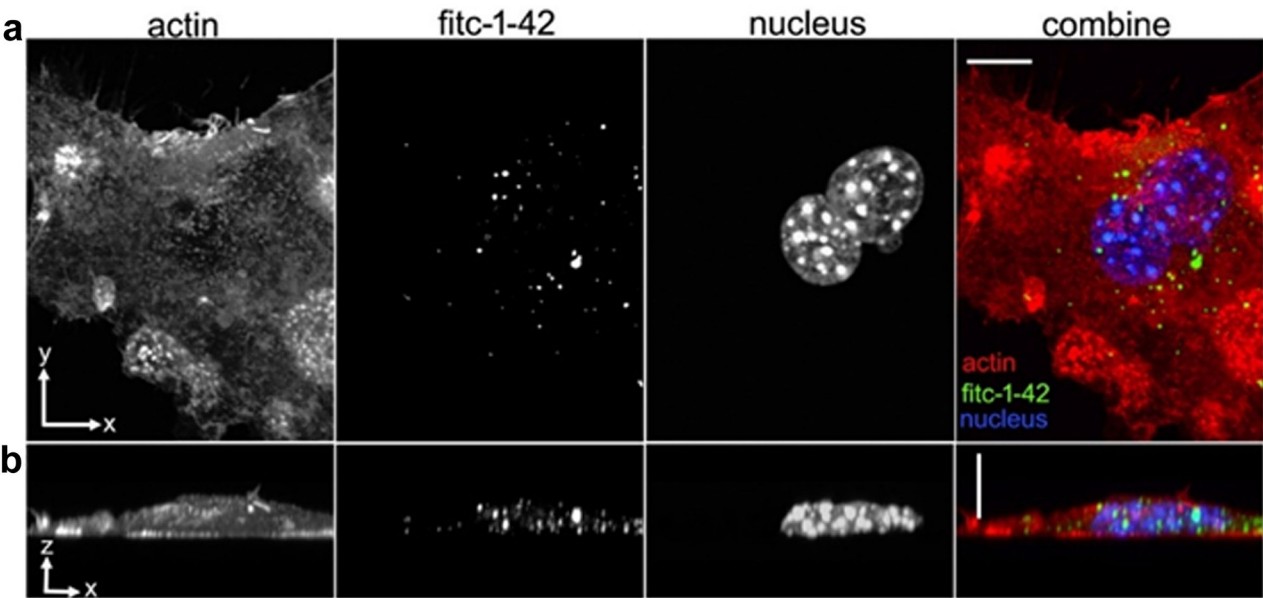

**Fig 6. BV2 microglial cell phagocytosis of Aβ1–42.** Non-stimulated BV2 cells were incubated with 100 nM FITC-Aβ1–42 for 24 hours, fixed and then stained for actin and nuclei. **(A)** The x-y and **(B)** x-z views of individual and combined actin, FITC-Aβ1–42 and nuclei channels. The FITC-Aβ1–42 localizes perinuclear and throughout the cell thickness. Scale bars = 10 microns.

other studies [27], NNC did not have any impact. Yet, SRIF treatment has previously shown to inhibit microglia viability relative to proliferation rate [28]. As SRIF binds all SSTR subtypes, our data would support a differential SSTR subtype LPS response in microglia where $SSTR_4$ may not induce an anti-proliferative effect. Moreover, under non-LPS treatment, increasing concentrations of NNC reduced LDH in the media. As LDH is a cytoplasmic protein, its relative decrease would support a maintenance of microglia cell membrane integrity potentially through $SSTR_4$ activation.

The LPS-induced NO was inhibited by NNC treatment, which aligns with previous work identifying the ability of SRIF to inhibit iNOS expression during LPS co-administration [29]. NO release from microglia under inflammatory conditions can enhance oxidative stress directly or through induction of downstream reactive species. Under pathological conditions NO contributes to axonal and synaptic damage, inhibition of mitochondrial respiration, and cell death [30]. Moreover, NO is linked to Aβ aggregation and NO can contribute to protein tyrosine nitration through secondary products like nitrogen dioxide and peroxynitrite. Nitration of $Aβ_{1-42}$ at tyrosine-10 has shown to enhance aggregation and localization to the core plaques of human AD brains and APP/PS1 mice [31]. $Aβ_{1-42}$ further demonstrates the tendency to form higher molecular weight oligomers with peroxynitrite treatment *in vitro* [31]. Inversely, NNC treatment in both Senescence Accelerated Mouse-Prone 8 (SAMP8) and APPswe mouse models has shown to decrease higher molecular weight $Aβ_{1-42}$ oligomer protein expression in cortical tissue [7, 8]. While the action of Aβ-degrading enzymes were identified as the cause of the decreased brain $Aβ_{1-42}$ oligomer expression with NNC treatment respective to the SAMP8 and APPswe studies, a reduced microglia NO output under proinflammatory conditions may afford an additional advantage.

In the 96-well plate assays (alamar blue, nitrite, and lactate dehydrogenase), we explored a range of concentrations (0, 0.104, 0.208, 0.417, 0.833 μM). The NNC compound had a maximal effect in the nitrite assay at 0.833 μM, which decreased LPS-induced nitrite to near

baseline. The flow cytometry-based assays (sytox, cell scattering, cell calcium and $A\beta_{1-42}$ uptake) are sensitive but low throughput, so we choose just one concentration to forward. We choose 1.0 μM, a concentration marginally above 0.833 μM, and likely to show maximal effect without cell toxicity.

We observed a sustained increased in intracellular calcium levels after stimulation with LPS for 24 hr, which is consistent with findings in primary microglia from mice. Chelation of extra-cellular calcium partially inhibited LPS-stimulated release of NO in microglia while artificially increasing intracellular calcium using an ionophore was not sufficient to cause NO release suggesting cellular activation requires intracellular calcium signaling [32]. In our experiments, NNC was able to decrease the LPS-stimulated calcium increases after 24 hrs indicating a mechanism through regulation of cytosolic calcium concentration. Aβ peptide, in addition to LPS, may cause a sustained increase in microglia calcium leading to cellular dysfunction [32, 33]. Through downregulation of calcium in microglia, NNC may block NO release and restore cellular function to maintain phagocytosis.

Assessment of sytox green showed only LPS significantly influenced cell viability, corresponding with the alamar blue data. Similarly, SSC and FSC also increased with LPS, affiliated with intracellular granularity and size, respectively. This corresponds with microglia activations shifting from a ramified state to a more amoeboid phenotype with large soma, corresponding to a state of enhanced phagocytic capacity. While SRIF has been shown to increase microglia phagocytosis of $A\beta_{1-42}$ [18], this is the first identification of a selective $SSTR_4$ agonist to increase microglia phagocytosis of monomeric $A\beta_{1-42}$. The effect was consistent over both the 24 hr and the abbreviated 2 hr FITC-$A\beta_{1-42}$ uptake assessments. However, LPS stimulation diminished the $A\beta_{1-42}$ uptake effect in exposure periods. This observation is corresponds with previous work evaluating SRIF treatment of microglia [18]. As elevated levels of LPS corresponded to increased Aβ phagocytosis the threshold of phagocytic capacity may well be reached within the assay, reducing the measurable impact of $SSTR_4$ activity. Nonetheless, the capacity of NNC to increase microglia $A\beta_{1-42}$ uptake in the absence of inflammatory activation may be particularly advantageous, given inflammatory activation is itself detrimental in AD populations [34]. This also aligns with our previous *in vivo* work with 3xTg mice, showing enhanced mRNA expression of key microglia proteins affiliated with Aβ phagocytosis and degradation under non-inflammatory conditions [10].

In conclusion, NNC 26–9100 decreased NO and cytosolic calcium levels under inflammatory conditions, while increasing $A\beta_{1-42}$ uptake under non-inflammatory conditions. Given the complexity of microglia contributions in AD, further evaluations are still needed to delineate the potential full intracellular responses of $SSTR_4$ activation in human AD tissues. Nevertheless, the current observations in conjunction with other work identifying anti-inflammatory capacity [29, 35, 36], cognitive enhancement [7–9], and the ability to decrease cortical Aβ oligomer expression [7, 8] supports further investigation of $SSTR_4$ agonists for AD treatment.

## Author Contributions

**Conceptualization:** Joseph Schober, Ken Witt.

**Data curation:** Karin Sandoval.

**Formal analysis:** Karin Sandoval.

**Funding acquisition:** Ken Witt.

**Investigation:** Joseph Schober, Jahnavi Polina, Field Walters, Nathan Scott, Eric Lodholz, Ken Witt.

**Methodology:** Jahnavi Polina, Field Walters, Nathan Scott, Eric Lodholz.

**Resources:** Albert Crider, Ken Witt.

**Supervision:** Joseph Schober, Ken Witt.

**Writing – original draft:** Karin Sandoval, Ken Witt.

**Writing – review & editing:** Joseph Schober, Karin Sandoval, Ken Witt.

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
