## [Decision Letter · Decision Letter 0]

30 May 2021

PONE-D-21-15019

NNC 26-9100 Increases Aβ1-42 Phagocytosis, Inhibits Nitric Oxide Production and Decreases Calcium in BV2 Microglia Cells

PLOS ONE

Dear Dr. Schober,

Thank you for submitting your manuscript to PLOS ONE. After careful consideration, we feel that it has merit but does not fully meet PLOS ONE’s publication criteria as it currently stands. Therefore, we invite you to submit a revised version of the manuscript that addresses the points raised during the review process.

Two experts have seen the manuscript and only asked for minor changes, as detailed in their reviews. Modifications should include justification for the agonist concentrations used in different experiments, addition of abbreviations and further explanations regarding the therapeutic importance of the study.

We look forward to receiving your revised manuscript.

Kind regards,

Mária A. Deli, M.D., Ph.D.

Academic Editor

PLOS ONE

Journal Requirements:

3. In your Methods section, please provide additional details regarding the cell lines used in your study and ensure you have described the source. For more information regarding PLOS' policy on materials sharing and reporting, see https://journals.plos.org/plosone/s/materials-and-software-sharing#loc-sharing-materials, and for more information on PLOS ONE's guidelines for research using cell lines, see https://journals.plos.org/plosone/s/submission-guidelines#loc-cell-lines.

4. We note that you have a patent relating to material pertinent to this article. Please provide an amended statement of Competing Interests to declare this patent (with details including name and number), along with any other relevant declarations relating to employment, consultancy, patents, products in development or modified products etc. Please confirm that this does not alter your adherence to all PLOS ONE policies on sharing data and materials, as detailed online in our guide for authors http://journals.plos.org/plosone/s/competing-interests by including the following statement: "This does not alter our adherence to  PLOS ONE policies on sharing data and materials.” If there are restrictions on sharing of data and/or materials, please state these. Please note that we cannot proceed with consideration of your article until this information has been declared.

Authors K. Witt, A. Crider, and K. Sandoval are co-inventors on patent applications for the development of SSTR4 agonists, through Southern Illinois University Edwardsville (USA patent application No. PCT/US2018/032368; European National Application No. 18798793.8, Japanese Application No. 2019-562574). Ken Witt is CEO of Somatolynk Inc, a company focused on advancement of SST4 agonists.

We note that one or more of the authors are employed by a commercial company: Somatolynk Inc

The National Institute on Aging of the National Institutes of Health award number R01AG047858 and SIUE Research Grants for Graduate Students supported this research.

"KW is supported by the National Institute on Aging of the National Institutes of Health grant R01AG047858."

Reviewers' comments:

Reviewer's Responses to Questions

**Comments to the Author**

1. Is the manuscript technically sound, and do the data support the conclusions?

Reviewer #1: Yes

Reviewer #2: Yes

2. Has the statistical analysis been performed appropriately and rigorously? 

Reviewer #1: Yes

Reviewer #2: Yes

3. Have the authors made all data underlying the findings in their manuscript fully available?

Reviewer #1: Yes

Reviewer #2: Yes

4. Is the manuscript presented in an intelligible fashion and written in standard English?

Reviewer #1: Yes

Reviewer #2: Yes

5. Review Comments to the Author

Reviewer #1: The manuscript examines the effect on SST4 receptor agonist NNC 26-9100 on BV2 microglia cells with special focus on cell function related to Alzheimer’s disease. The design of the study is elaborate. Methods are modern and English language is excellent. I have two minor questions:

Why is the maximal concentration of NNC 26-9100 different in Alamar blue, LDH and nitrite assays compared to the other experiments?

They authors conclude on shift of microglia to M2 phenotype. This should only be included if respective cell markers are examined and phenotype shift can be objectively demonstrated.

Reviewer #2: The well-written manuscript of Schober and co-authors presents a nice and thorough work, analysing the effects of an SSTR4 agonist on microglia function. The Authors confirmed that NNC Is able to decrease the LPS-induced lactate dehydrogenase and nitrite release, while in the absence of LPS, increase Aβ uptake. The data presented in this manuscript confirm the previous results of the Authors. In my opinion the results are clear and well-described, only few modifications are needed:

The explanations of some abbreviations are missing from the manuscript (SSTR4, SRIF).

The article would greatly benefit from a brief description of the role of SSRTs and SRIF in AD, to emphasize the potential therapeutic applicability of the results.

6. PLOS authors have the option to publish the peer review history of their article (what does this mean?). If published, this will include your full peer review and any attached files.

Reviewer #1: No

Reviewer #2: No

---

## [Author Response · Author response to Decision Letter 0]

17 Jun 2021

We thank the Reviewers for their comments. Each of the text changes in the manuscript are highlighted by underlines. 

Responses to Reviewers

Reviewer #1: The manuscript examines the effect on SST4 receptor agonist NNC 26-9100 on BV2 microglia cells with special focus on cell function related to Alzheimer’s disease. The design of the study is elaborate. Methods are modern and English language is excellent. I have two minor questions:

Why is the maximal concentration of NNC 26-9100 different in Alamar blue, LDH and nitrite assays compared to the other experiments?

Author Response- In the 96-well plate assays (alamar blue, nitrite, and lactate dehydrogenase), we explored a range of concentrations (0, 0.104, 0.208, 0.417, 0.833 �M). The NNC compound had a maximal effect in the nitrite assay at 0.833 �M, which decreased LPS-induced nitrite to near baseline. The flow cytometry-based assays (sytox, cell scattering, cell calcium and Aβ1-42 uptake) are sensitive but low throughput, so we choose just one concentration to forward. We choose 1.0 �M, a concentration marginally above 0.833 �M, and likely to show maximal effect without cell toxicity. We included this text in the Discussion section of the revised manuscript. 

They authors conclude on shift of microglia to M2 phenotype. This should only be included if respective cell markers are examined and phenotype shift can be objectively demonstrated. 

Author Response- We agree with the reviewer. The sentence referring to M2 phenotype in the last paragraph in the Discussion was removed. 

Reviewer #2: The well-written manuscript of Schober and co-authors presents a nice and thorough work, analyzing the effects of an SSTR4 agonist on microglia function. The Authors confirmed that NNC is able to decrease the LPS-induced lactate dehydrogenase and nitrite release, while in the absence of LPS, increase Aβ uptake. The data presented in this manuscript confirm the previous results of the Authors. In my opinion, the results are clear and well-described, only few modifications are needed:

The explanations of some abbreviations are missing from the manuscript (SSTR4, SRIF).

Author Response- The abbreviations are now defined at their first usage. 

The article would greatly benefit from a brief description of the role of SSRTs and SRIF in AD, to emphasize the potential therapeutic applicability of the results.

Author Response- A section in the Discussion beginning was added which points the reader to references on the topic of somatostatin and somatostatin receptors in AD.

---

## [Decision Letter · Decision Letter 1]

23 Jun 2021

NNC 26-9100 Increases Aβ1-42 Phagocytosis, Inhibits Nitric Oxide Production and Decreases Calcium in BV2 Microglia Cells

PONE-D-21-15019R1

Dear Dr. Schober,

We’re pleased to inform you that your manuscript has been judged scientifically suitable for publication and will be formally accepted for publication once it meets all outstanding technical requirements.

Kind regards,

Mária A. Deli, M.D., Ph.D.

Academic Editor

PLOS ONE

Additional Editor Comments (optional):

Reviewers' comments:

Reviewer's Responses to Questions

**Comments to the Author**

1. If the authors have adequately addressed your comments raised in a previous round of review and you feel that this manuscript is now acceptable for publication, you may indicate that here to bypass the “Comments to the Author” section, enter your conflict of interest statement in the “Confidential to Editor” section, and submit your "Accept" recommendation.

Reviewer #1: All comments have been addressed

Reviewer #2: All comments have been addressed

2. Is the manuscript technically sound, and do the data support the conclusions?

Reviewer #1: Yes

Reviewer #2: (No Response)

3. Has the statistical analysis been performed appropriately and rigorously? 

Reviewer #1: Yes

Reviewer #2: (No Response)

4. Have the authors made all data underlying the findings in their manuscript fully available?

Reviewer #1: Yes

Reviewer #2: (No Response)

5. Is the manuscript presented in an intelligible fashion and written in standard English?

Reviewer #1: Yes

Reviewer #2: (No Response)

6. Review Comments to the Author

Reviewer #1: The authors answered all of my questions. The already high quality paper has been corrected even further. The paper is now suitable for publication.

Reviewer #2: (No Response)

7. PLOS authors have the option to publish the peer review history of their article (what does this mean?). If published, this will include your full peer review and any attached files.

Reviewer #1: **Yes: **Dr. Gabor Pozsgai

Reviewer #2: No

---

## [Editor Report · Acceptance letter]

28 Jun 2021

PONE-D-21-15019R1 

NNC 26-9100 Increases Aβ1-42 Phagocytosis, Inhibits Nitric Oxide Production and Decreases Calcium in BV2 Microglia Cells 

Dear Dr. Schober:

I'm pleased to inform you that your manuscript has been deemed suitable for publication in PLOS ONE. Congratulations! Your manuscript is now with our production department. 

Kind regards, 

on behalf of

Dr. Mária A. Deli 

Academic Editor

PLOS ONE